# What drives willingness to receive a new vaccine that prevents an emerging infectious disease? A discrete choice experiment among university students in Uganda

Kimberly E. Bonner[1]*, Henry Ssekyanzi[2], Jonathan Sicsic[3], Judith E. Mueller[4,5], Traci Toomey[1], Angela K. Ulrich[1], Keith J. Horvath[6], James D. Neaton[7], Cecily Banura[8‡], Nicole E. Basta[9‡]

1 Division of Epidemiology and Community Health, School of Public Health, University of Minnesota, Minneapolis, Minnesota, United States of America, 2 College of Health Sciences, Makerere University, Kampala, Uganda, 3 University of Paris, LIRAES, Paris, France, 4 EHESP French School of Public Health, La Plaine St Denis, France, 5 Institute Pasteur, Paris, France, 6 Department of Psychology, San Diego State University, United States of America, 7 Division of Biostatistics, School of Public Health, University of Minnesota, Minneapolis, MN, United States of America, 8 Child Health and Development Centre, College of Health Sciences, School of Medicine, Makerere University, Kampala, Uganda, 9 Department of Epidemiology, Biostatistics, and Occupational Health, School of Population and Global Health, Faculty of Medicine, McGill University, Montreal, Quebec, Canada

‡ These senior co-authors contributed equally to the work.
* Kimberly.bonner@gmail.com

## Abstract

### Background

There is a critical need to identify the drivers of willingness to receive new vaccines against emerging and epidemic diseases. A discrete choice experiment is the ideal approach to evaluating how individuals weigh multiple attributes simultaneously. We assessed the degree to which six attributes were associated with willingness to be vaccinated among university students in Uganda.

### Methods

We conducted a single-profile discrete choice experiment at Makerere University in 2019. Participants were asked whether or not they would be vaccinated in 8 unique scenarios where attributes varied by disease risk, disease severity, advice for or against vaccination from trusted individuals, recommendations from influential figures, whether the vaccine induced indirect protection, and side effects. We calculated predicted probabilities of vaccination willingness using mixed logistic regression models, comparing health professional students with all other disciplines.

### Findings

Of the 1576 participants, 783 (49.8%) were health professional students and 685 (43.5%) were female. Vaccination willingness was high (78%), and higher among health students

data, given that these were the specifications listed in the protocols reviewed by the ethics committees at the School of Medicine Research Ethics Committee (SOMREC) at Makerere University, the University of Minnesota Institutional Review Board (IRB), and the Uganda National Council for Science and Technology (UNCST). Data requests can be sent to Molly McCoy, the Assistant Director for Compliance for International Health, Safety, and Compliance Global Programs and Strategy Alliance at the University of Minnesota via the email address "mccoy019@umn.edu."

**Funding:** NPGH Fogarty Global Health Fellowship; the Consortium for Law and Values in Health, Environment and the Life Sciences, University of Minnesota.

**Competing interests:** The authors have declared that no competing interests exist.

than other students. We observed the highest vaccination willingness for the most severe disease outcomes and the greatest exposure risks, along with the Minister of Health's recommendation or a vaccine that extended secondary protection to others. Mild side effects and recommendations against vaccination diminished vaccination willingness.

## Interpretation

Our results can be used to develop evidence-based messaging to encourage uptake for new vaccines. Future vaccination campaigns, such as for COVID-19 vaccines in development, should consider acknowledging individual risk of exposure and disease severity and incorporate recommendations from key health leaders.

## Introduction

Emerging and epidemic infections pose a unique and urgent challenge in an increasingly interconnected world, as has been evidence by the ongoing COVID-19 pandemic, multiple recent and ongoing Ebola outbreaks, but also dozens of episodes of newly emerging or re-emerging diseases each year [1]. Many vaccines are known not only to protect individuals from disease but also slow the spread of disease in the community. Although vaccine development usually takes more than a decade several more recent examples have demonstrated the feasibility and impact of rapidly developing and deploying vaccines to reduce epidemic morbidity and mortality [2–4].

However, the impact of any newly developed vaccine depends on the proportion of individuals who express willingness to be vaccinated and seek out vaccination. Willingness is defined as having an intent or motivation to be vaccinated and is used as a proxy for vaccination uptake when assessing views on a novel vaccine not yet available. The World Health Organization (WHO) Increasing Vaccination model notes that vaccination motivation incorporates vaccination willingness and is influenced by what people think and feel, social processes, and practical issues [5]. This model incorporates the health belief model, including risk appraisal and vaccine confidence, into the thinking and feeling domain, while adding additional domains, including social process, and practical issues [6,7]. Vaccine hesitancy, defined as a delay in acceptance or refusal of a vaccine, results from a lack of confidence in vaccines, complacency towards vaccines, and inconvenience in accessing a vaccine [8] and can be modulated by factors such as interest in collective protection [9] and social conformism.

In the context of an emerging epidemic disease, specific drivers of vaccination willingness have not been fully elucidated and few studies that have addressed this question rigorously. However, previous studies have identified considerations that include the epidemiology of the disease, sources of information in support of or against vaccination, and characteristics of the vaccine itself [10–12]. During the 2014–6 Ebola outbreak in West Africa, a survey of adults found that acceptability of the Ebola vaccine was high (72.5%-80%) in Nigeria and Sierra Leone [13,14] but lower (34%) for adults in the United States [15,16]. Vaccine communication, including vaccination promotion messages from public health authorities or rumors, can influence willingness to receive a new vaccine [11,17]. Vaccines that extended protection beyond the individual were preferred by young adults in France [18]. Additionally, the risk of side effects may also influence willingness [10,19].

Despite the growing body of prior research on vaccination willingness, there is a gap in understanding how multiple competing factors influence vaccination willingness, especially in

resource-limited settings. Global vaccine hesitancy surveys have measured self-reported hesitancy [20], but not the underlying factors that contribute to vaccine hesitancy. In a survey of immunization managers in thirteen countries, the highest rate of missed vaccination opportunities due to vaccine hesitancy occurred in Uganda [21]. Given the reports of vaccine hesitancy, coupled with the frequent emergence of viral haemorrhagic fevers in Uganda [22], it is an important setting to examine the drivers of vaccination willingness.

Health professionals play a critical role in promoting vaccination. Therefore, there is a need to understand what motivates health professionals' vaccination willingness, as their personal vaccine willingness is tied to their recommendations to patients [23,24]. Of concern, a prior study in Liberia during the 2014–16 EVD epidemic found low willingness to receive a newly developed Ebola vaccine among healthcare workers [25]. Additionally, there is a need to understand the willingness of those who have newly entered or will soon enter the health workforce, as these health professionals can influence patients throughout the course of their careers. Here we define health fields as those in which students might themselves administer vaccines to patients or advise patients on matters of vaccination.

There is a need to understand the drivers of vaccination willingness, but many surveys are not structured to allow individuals to simultaneously weigh the multiple factors that drive vaccination willingness. Identifying these driving factors is particularly important for vaccines against emerging diseases as there is a limited time window to distribute vaccines to those who need them most. Because vaccine willingness may be context-dependent, there is a need to examine key populations, such as health care workers, and in countries like Uganda where hesitancy has been identified as a challenge and further research is needed to understand what motivates willingness in this context.

To examine the parameters that influence vaccination willingness, we used a discrete choice experiment (DCE) survey. A DCE is a study design that models the complexity of decision-making by allowing respondents to weigh different parameters simultaneously, rather than identifying a single parameter, a limitation of traditional surveys. DCEs measure the dominant driver of a single decision when there are multiple competing factors. Some concepts of behavioral economics, such as discounting, are explicit in DCEs only if economic considerations are central to the aims of the study [26,27] but not in a DCE on willingness to accept a vaccine. DCEs have quantified preferences in vaccine product profiles [28,29] and vaccine communication [30]. DCEs conducted in Europe have identified physicians' recommendation as the most influential driver of vaccination willingness against a pandemic disease [10,31]. A DCE in France identified the epidemic context and the vaccine's secondary protection as the key drivers, while controversy around its safety were the major detractor [18]. To our knowledge, only one DCE has examined vaccination preferences in sub-Saharan Africa, finding that vaccine effectiveness and accessibility were the key drivers of willingness for an unnamed vaccine for the population [19]. However, no studies have examined drivers of vaccine willingness for a new vaccine against an emerging disease in a low-income country, examining epidemic context, communication regarding the vaccine, and vaccine characteristics simultaneously. Furthermore, no studies have examined if these drivers differ between those training to become health professionals compared to those who are not receiving this training.

The purpose of this study is to examine the role of multiple factors that could influence an individual's willingness to be vaccinated with a new vaccine that protects against an emerging epidemic disease among young adults in Uganda. We had three primary aims: (1) to estimate and compare willingness to be vaccinated among university students in health fields (those enrolled in the College of Health Sciences) and students enrolled in other fields; (2) to identify which parameters drive willingness to be vaccinated overall and among students in health fields and all others disciplines; and (3) to compare the magnitude of the association between

those parameters identified in Aim 2 and willingness to be vaccinated overall and among students in health fields and all others disciplines. By accomplishing these aims, we will be able to describe willingness to be vaccinated with a new vaccine for an emerging disease among university students, to identify which factors drive this willingness, and to assess the relative strength of those factors in driving willingness to be vaccinated. Our goal is to identify factors that are most strongly associated with willingness so that future vaccination campaigns can address these factors when new vaccines are introduced.

## Methods

### Study design

We conducted a DCE survey among Makerere University students in Kampala, Uganda between February 13th-March 16th, 2019. Eligible students were aged 18 years and above, able to read and speak English (the national language of Uganda), and were current students in one of the six largest Colleges at Makerere University (1. Business and Management Sciences, 2. Computing and Information Science, 3. Education and External Studies, 4. Health Science, 5. Humanities and Social Science, and 6. Veterinary Medicine, Animal Resources, and Biosecurity). We define students in health fields as those enrolled in the College of Health Sciences. Pilot testing occurred at the College of Veterinary Medicine prior to the launch of the study. For the survey, a convenience sample was recruited through posters and WhatsApp messages sent by student leaders. Participants were enrolled from all six colleges across five enrollment sites.

Eligible students interested in participating provided informed verbal consent. The School of Medicine Research Ethics Committee (SOMREC) at Makerere University, the University of Minnesota Institutional Review Board (IRB), and the Uganda National Council for Science and Technology (UNCST) provided ethical approval for this study.

We aimed to recruit a minimum of 575 and maximum of 800 participants from two groups of students: health sciences and other disciplines, which enables us to detect up to a true seven-percentage point difference plus or minus two points between groups with an alpha 0.05 and 80% power. While no definitive statistical method has been established for DCE sample size calculations, this sample size exceeds Orme's commonly-used calculation for DCEs [32].

### Survey development

The 21-question survey included six demographic questions, five questions on vaccination attitudes and vaccination history, nine discrete choice experiment questions (Fig 1), and one question for a sensitivity analysis (S1 Table).

The survey was pilot tested with 15 students selected from Makerere University using a thinking aloud exercise to identify preferences for survey administration and refine the questions, a standard practice for DCEs [33–36]. Following the pilot test, we drafted a standard script for study staff to use in an example question with each participant.

The survey collected responses anonymously using the Qualtrics platform [37] and was administered using a study tablet at the study sites. Students were given a choice to self-administer the survey or to have study staff administer it in the first two data collection sites. The survey was administered by study staff for the final three sites. All students received UGX 10,000 (~USD $2.80) along with a soda and chocolate as compensation for the 20-minute survey.

We developed the DCE survey tool in accordance with the International Society for Pharmacoeconomics Outcomes Research (ISPOR) guidelines [33]. We undertook a literature review of other DCEs on adult vaccination [10,18,19,31,38,39] and the broader literature on vaccine willingness, which identified six key attributes that could influence vaccination

There is an outbreak of a disease that is usually rare

- The disease can kill you
- You can catch the disease by close contact with a sick person (either by touching them or their body fluids)
- There have been less than 50 cases reported so far in East Africa

The Ministry of Health provides a new vaccine against the disease free of charge.

- 7 out of 10 people who get the vaccine will be fully protected from the disease
- The vaccine is one injection
- The vaccine has not yet received full regulatory approval
- The vaccine has never been used before in Uganda

For the next nine questions, decide **whether or not you would take the vaccine if it were offered to you.** Next, decide **how much you would want the vaccine** on a scale of 1-10, with 10 being the most. *The characteristics of the disease and the vaccine will change slightly in each question, so please read carefully to see the difference.*

Decide if you would take this new vaccine in Scenario CC.

**Disease Risk:** Someone in your **current household** touched an infected person
**Disease Severity:** The disease kills **1% (1 in 100)** of people infected
**Trusted Individuals:** A **family member or friend** that you trust advised you **to take** the vaccine
**Influential Voices:** The **Minister of Health** recommended that people **take** the vaccine
**Vaccine Protection:** By getting vaccinated, you protect yourself **and others**
**Vaccine Side Effects:** The vaccine gives **20% (2 in 10)** of people **a high fever** for 1 day

Would you choose to receive the vaccine?

Yes

No

**Fig 1. Example DCE scenario and DCE question.**

willingness in the context of an emerging disease. We listed the attributed identified by the literature review and selected the six attributes with the greatest magnitude of effect. The **attributes** that we investigated in this study are: 1) the degree of **disease risk** (varying based on proximity to a case); 2) the degree of **disease severity** ranging from a 50% fatality rate to a 0.1% fatality rate; 3) the advice of **trusted individuals**, both positive and negative; 4) the advice of **influential voices** from leaders, both positive and negative; 5) the nature of **vaccine protection** (whether the vaccine provided indirect protection to the community or not); and 6) information about the nature and degree of **side effects** (Table 1). For each attribute, we set the reference as the least likely to be associated with vaccination willingness.

We used a fractional factorial design to efficiently select 32 out of the 1,062 possible combinations of attribute-specific parameters with four blocks of eight randomly ordered questions using SAS Optex [40]. In each survey, a participant was asked one duplicate DCE question as a consistency check; these were always presented first and last among the scenarios. Before each DCE question, a framing situation was presented describing the overall context (Fig 1).

**Table 1. Attributes and attribute-specific parameters for a new vaccine against an emerging disease[t].**

| Attribute | Attribute-specific parameter |
| --- | --- |
| Disease Risk | 1. Someone in your current household touched an infected person |
|  | 2. cases of the disease were just reported in your district |
|  | 3. cases of the disease were just reported in a distant region of Uganda |
|  | 4. cases of the disease were reported in a neighboring country (ref) |
| Disease Severity | 1. The disease kills 50% (5 in 10) of people infected |
|  | 2. The disease kills 10% (1 in 10) of people infected. |
|  | 3. The disease kills 1% (1 in 100) of people infected |
|  | 4. The disease kills 0.1% (1 in 1,000) of people infected (ref) |
| Trusted Individuals | 1. A family member or friend that you trust advised you to take the vaccine |
|  | 2. A religious or tribal leader that you trust advised you to take the vaccine |
|  | 3. A religious or tribal leader that you trust advised you not to take the vaccine |
|  | 4. A family member or a friend that you trust advised you not to take the vaccine (ref) |
| Influential Voices | 1. The Minister of Health recommended that people take the vaccine |
|  | 2. Your favorite social media blogger advised people to take the vaccine<br>3. Your favorite social media blogger advised people not to take the vaccine |
|  | 4. An opposition politician warned people not to take the vaccine(ref) |
| Vaccine Protection | 1. By getting vaccinated, you protect yourself and others |
|  | 2. By getting vaccinated you protect only yourself, but not others (ref) |
| Side Effects | 1. The vaccine gives 20% (2 in 10) of people a skin rash somewhere on their body for 3 days |
|  | 2. The vaccine gives 20% (2 in 10) of people a high fever for 1 day<br>3. You've heard rumors about harmful side effects, but none have been confirmed |
|  | 4. The vaccine injection is painful for 30 minutes (ref) |

[t]For each DCE question, each student received a scenario displaying one parameter per attribute and was asked whether or not they would choose to receive the vaccine given the combination of those six attributes presented together in that scenario. (ref) indicates the reference parameter level used in analysis to compare to the other levels of that parameter.

## Measurement

Our outcome of interest was whether a student would be willing (i.e. willingness) to receive the vaccine (binary; 1 = willing to be vaccinated, 0 = not willing), given the combination of six attribute-specific parameters presented in each question. The independent variables of interest were the six different attributes listed in Table 1. Other covariates were identified *a priori*: sex (male vs female); age (continuous); region of birth (Central, Western, Northern, Eastern, born outside Uganda, or do not know/not sure); religion (Catholic, Muslim, Pentecostal, Protestant, or other); and Hepatitis B vaccination history (yes vs no/do not know/do not remember).

## Statistical analyses

To estimate and compare vaccination willingness between students in health disciplines to those in other disciplines, we used mixed logistic regression model (xtlogit) with a random intercept to ascertain vaccination willingness of the population average effects for each group. The dependent variable was vaccination willingness. Because each participant received eight unique scenarios and indicated their vaccine willingness eight times, their responses formed a panel, clustered at the individual level. Models were adjusted for sex, age; region of birth, religion, and Hepatitis B vaccination history. From the model outputs, we used the margins command to calculate the predicted probability and 95% confidence interval (CI) of willingness in each group and the difference in willingness between groups.

To estimate the magnitude of the association between each of the attribute-specific parameters (attributes: disease risk (four parameters), disease severity (four parameters), trusted individuals (four parameters), influential voices (four parameters), vaccine protection (two parameters), and side effects (four parameters) and the outcome of vaccination willingness, we used mixed logistic regression models to account for individual clustering across responses for all students and models stratified by discipline of study. We reported the odds ratios and 95% CIs for the relationship between each of the parameters (compared to the reference parameter for a given attribute category) and vaccination willingness.

To estimate the predicted probability of vaccination willingness for each of the attribute-specific parameters for both students in health disciplines and other disciplines, we included health discipline as an interaction term in a mixed logistic regression model (xtlogit) including all students, adjusting for all covariates listed above and accounting for individual-level clustering. Using marginsplot, we graphed the predicted probability and 95% CIs of willingness for the average individual within each group, given the presence of each parameter. For example, the predicted probability that a health professional student would be willing to be vaccinated if the vaccine were recommended by the Minister of Health, given that all other covariates are held at their weighted distribution in the study population.

Sensitivity analyses: We undertook an assessment of data quality by conducting four sensitivity analyses [S1 & S2 Tables] [41]. We created four subgroups by: 1) survey time: surveys completed in eight minutes or longer; 2) duplicates: surveys where the duplicate questions were answered consistently; and 3) attributes considered, with surveys where at least half of the six attributes were "always" or "often" considered. A fourth sensitivity analysis examined whether the 4) route of survey administration (self or interviewer-administered) affected the predicted probability of answering the duplicate questions consistently.

Stata 16 was used for all analyses [42].

## Role of the funding source

Study sponsors had no role in the study design; in the collection, analysis, and interpretation of data; in the writing of the report; and in the decision to submit the paper for publication.

## Results

### Descriptive statistics

Overall, 1600 students participated in the study, including 800 students in health disciplines and 800 students from other disciplines. Of these, 24 surveys did not have appropriate information to be included in the analysis due to incomplete surveys, misassigned ID number, or inability to locate staff documentation of their verbal consent. The final analytic dataset consisted of 1576 participants; 783 students in health disciplines and 793 students from other disciplines. The participant demographics are listed in Table 2.

The overall predicted proportion willing to receive a newly-developed vaccine in the context of an emerging epidemic was 78.0% (95% CI, 76.8%-79.2%), with a higher proportion of health students willing to be vaccinated compared to students in other disciplines. (82.1% vs. 74.0%) (Table 3).

The odds of willingness to receive a vaccine was higher for most parameters within an attribute (Table 4). At higher levels of **disease risk**, the odds of vaccination willingness were higher; we observed a 3.0-fold (95% CI, 2.6–3.5) higher willingness if a household member had direct contact with a case, compared to a situation where cases were detected in a neighboring country. Similarly, at higher levels of **disease severity** defined by high case fatality, the odds of vaccination willingness were up to 5.0-fold higher (95% CI 4.2–5.8) compared to the lowest

**Table 2. Demographic characteristics of study participants who completed the DCE to assess their willingness to receive a vaccine.**

| | | Health Disciplines N = 783 | Other disciplines N = 793 |
|---|---|---|---|
| | | N (%) | N (%) |
| Age (Mean(SD)) | | 24 (4) | 22 (2) |
| Sex | Female | 276 (35) | 409 (52) |
| | Male | 507 (65) | 384 (49) |
| Birth Region | Central Uganda | 358 (46) | 371 (47) |
| | Western Uganda | 166 (21) | 246 (31) |
| | Northern Uganda | 68 (9) | 33 (4) |
| | Eastern Uganda | 150 (19) | 134 (17) |
| | Born outside of Uganda | 41 (5) | 7 (1) |
| | Don't know/Not sure | 0 (0) | 2 (0) |
| Religion | Catholic | 240 (31) | 259 (33) |
| | Muslim | 76 (10) | 76 (10) |
| | Pentecostal | 123 (16) | 80 (10) |
| | Protestant | 258 (33) | 324 (41) |
| | Other | 86 (11) | 54 (7) |
| Received Hepatitis B Vaccine | No/Don't know/Don't remember | 195 (25) | 503 (63) |
| | Yes | 588 (75) | 290 (37) |

reference case fatality presented; this was observed in both health students and non-health students. A positive recommendation from a family member or friend to take the vaccine was associated with a 1.9-fold higher (95% CI 1.6–2.2) vaccination willingness compared to receiving negative advice from a family member or friend to *not* take the vaccine. With the **influential voices** attribute, the Minister of Health's recommendation to receive the vaccine was associated with 1.8-fold (95% CI 1.6–2.2) higher vaccine willingness compared to an opposition politician's recommendation not to get vaccinated. A vaccine that induced herd immunity was associated with a 1.6-fold (95% CI 1.4–1.8) higher vaccination willingness compared to a vaccine that only protected the person being vaccinated. With regards to **side effects**, the risk of skin rash lasting three days was associated with a 0.6-fold lower vaccination willingness compared to the referent minimal side effect of 30 minutes of pain following injection. Rumors of harmful side effects were also associated with lower vaccination willingness compared to the referent parameter. The individual characteristics associated with vaccination willingness

**Table 3. Willingness to receive a new vaccine by college status, population average effects[*].**

| | Number (n) | Vaccine willingness predicted proportion % (95% CI) | Predicted proportion difference in vaccine willingness (95% CI) | P>\|z\| |
|---|---|---|---|---|
| Overall | 1,576 | 78.0% (76.8–79.2) [t] | - | - |
| Health disciplines | 783 | 82.1 (80.5–83.7) [Γ] | 6.7 (4.1–9.3)[*] | <0.01 |
| Other disciplines (ref) | 793 | 74.0 (72.3–75.6) [Γ] | | |

[*]Each model included covariates for sex, age, region of birth, religion, and Hepatitis B vaccination status.

[t]Mixed logistic regression with random intercepts.

[Γ]Mixed logistic regression with random intercepts, stratified by health or other discipline.

[*]Mixed logistic regression with random intercepts, including health or other discipline as a covariate. The predicted proportion of the difference in vaccine willingness setting all covariates at their mean distribution in the population.

**Table 4. Estimating the odds ratio (OR) of attribute-specific parameters associated with willingness to receive a new vaccine[t].**

| PARAMETERS | | All (adjusted) | Health disciplines adjusted | Other disciplines adjusted |
|---|---|---|---|---|
| | | OR (95% CI) | OR (95% CI) | OR (95% CI) |
| **Risk–Willingness to receive new vaccine given the following risk** | | | | |
| (highest risk) Someone in your current household touched an infected person | | 3.0 (2.6–3.5) | 3.5 (2.7–4.5) | 3.0 (2.4–3.6) |
| (higher risk) 2 cases of the disease were just reported in your district | | 1.5 (1.3–1.8) | 1.7 (1.3–2.1) | 1.5 (1.2–1.8) |
| (low risk) 2 cases of the disease were just reported in a distant region of Uganda | | 1.1 (0.9–1.2) | 1.0 (0.8–1.3) | 1.1 (0.9–1.4) |
| (lowest risk) 25 cases of the disease were just reported in a neighboring country | | ref | ref | ref |
| **Severity** | | | | |
| (highest risk) The disease kills 50% (5 in 10) of people infected | | 5.0 (4.2–5.8) | **8.9** (6.8–11.8) | **3.5** (2.9–4.3) |
| (high risk) The disease kills 10% (1 in 10) of people infected | | 2.4 (2.1–2.8) | 3.2 (2.5–4.1) | 2.0 (1.6–2.4) |
| (low risk) The disease kills 1% (1 in 100) of people infected | | 1.4 (1.3–1.7) | 1.7 (1.4–2.1) | 1.3 (1.0–1.5) |
| (lowest risk) The disease kills 0.1% (1 in 1,000) of people infected | | ref | ref | ref |
| **Trusted Individuals** | | | | |
| A family member or friend that you trust advised you to take the vaccine | | 1.9 (1.6–2.2) | 1.8 (1.4–2.3) | 1.9 (1.6–2.4) |
| A religious or tribal leader that you trust advised you to take the vaccine | | 1.6 (1.3–1.8) | 1.5 (1.2–1.9) | 1.6 (1.3–1.9) |
| A religious or tribal leader that you trust advised you not to take the vaccine | | 1.1 (0.9–1.3) | 1.1 (0.9–1.4) | 1.1 (0.9–1.3) |
| A family member or friend that you trust advised you not to take the vaccine | | ref | ref | ref |
| **Influential Voices** | | | | |
| The Minister of Health recommended that people take the vaccine | | 1.9 (1.6–2.2) | 1.8 (1.4–2.4) | 2.4 (2.0–3.0) |
| Your favorite social media blogger advised people to take the vaccine | | 1.6 (1.3–1.8) | 1.0 (0.8–1.3) | 1.3 (1.1–1.5) |
| Your favorite social media blogger advised people not to take the vaccine | | 1.1 (0.9–1.3) | 0.8 (0.6–1.0) | 0.9 (0.8–1.1) |
| An opposition politician warned people not to take the vaccine | | ref | ref | ref |
| **Vaccine Protection** | | | | |
| By getting vaccinated, you protect yourself and others | | 1.6 (1.4–1.8) | 1.5 (1.2–1.8) | 1.7 (1.4–1.9) |
| By getting vaccinated you protect only yourself, but not others | | ref | ref | ref |
| **Side Effects** | | | | |
| The vaccine gives 20% of people a skin rash somewhere on their body for 3 days | | 0.6 (0.5–0.7) | 1.0 (0.8–1.3) | 0.4 (0.3–0.5) |
| The vaccine gives 20% of people a high fever for 1 day | | 0.7 (0.6–0.9) | 1.1 (0.8–1.4) | 0.6 (0.5–0.7) |
| You've heard rumors about harmful side effects, but none have been confirmed | | 0.6 (0.5–0.7) | 0.7 (0.6–0.9) | 0.6 (0.5–0.7) |
| The vaccine injection is painful for 30 minutes | | ref | ref | ref |
| Log of the variance* | | 1.1 (1.0–1.2) | 1.4 (1.2–1.7) | 0.8 (0.6–1.0) |
| Sigma u | | 1.7 (1.6–1.9) | 2.1 (1.9–2.3) | 1.5 (1.3–1.6) |
| rho | | 0.5 (0.4–0.5) | 0.6 (0.5–0.6) | 0.4 (0.4–0.5) |
| Age | | 1.0 (1.0–1.1) | 1.0 (1.0–1.0) | 1.0 (1.0–1.1) |
| Sex | Female | 0.8 (0.6–1.0) | 0.7 (0.5–1.0) | 1.0 (0.7–1.2) |
| | Male | ref | ref | ref |
| Region of birth | Central | ref | ref | ref |
| | Western | 1.2 (1.0–1.6) | 1.2 (0.8–1.9) | 1.4 (1.0–1.8) |
| | Northern | 1.1 (0.7–1.7) | 1.1 (0.5–2.1) | 1.0 (0.5–1.9) |
| | Eastern | 1.6 (1.2–2.1) | 1.4 (0.8–2.2) | 1.9 (1.3–2.8) |
| | Outside Uganda | 0.6 (0.3–1.2) | 0.5 (0.2–1.1) | 0.5 (0.1–1.9) |

*(Continued)*

**Table 4.** (Continued)

| PARAMETERS | | All (adjusted) | Health disciplines adjusted | Other disciplines adjusted |
|---|---|---|---|---|
| Religion | Catholic | ref | ref | ref |
| | Muslim | 0.9 (0.6–1.4) | 0.9 (0.5–1.8) | 0.9 (0.6–1.5) |
| | Pentecostal | 1.4 (1.0–2.0) | 1.6 (0.9–2.9) | 1.2 (0.8–1.9) |
| | Protestant | 0.9 (0.7–1.2) | 1.0 (0.6–1.6) | 0.9 (0.7–1.2) |
| | Other | 0.8 (0.5–1.2) | 0.9 (0.5–1.7) | 0.7 (0.4–1.1) |
| Hepatitis B vaccine | No/Don't know/Don't remember | ref | ref | ref |
| | Yes | 2.0 (1.6–2.5) | 2.8 (1.9–4.3) | 1.2 (0.90–1.5) |
| Choice observations | | 12,608 | 6,264 | 6,344 |
| Number of participants | | 1,576 | 783 | 793 |

ᵗThis model used panel mixed logistic regression and include covariates for sex, age, region of birth, religion, and Hepatitis B vaccination status, stratified by students in health disciplines and students from other disciplines.

*Indicates the extent of individual-level variability, with a higher value indicating greater variability within individuals.

included receipt of hepatitis B vaccine which was associated with 2.8-fold (95% CI 1.9–4.3) higher odds of vaccination willingness, female sex (0.7, 95% CI 0.5–1.0) among health students; and Eastern region of birth (OR 1.9, 95% CI 1.3–2.8).

The predicted probability of vaccination willingness was highest (89.0% for students in health disciplines and 81.7% for students from other disciplines) at the highest **disease risk** (Fig 2, Panel 1). For the lowest disease risk (25 cases reported in a neighboring country), the predicted probability of willingness was 78.1% for health students and 69.3% for non-health students. At lower levels of **disease severity**, vaccination willingness was also lower (Panel 2). For each severity parameter, health students expressed a greater willingness than non-health students. Among the **trusted individuals** presented (Panel 3), the positive recommendation of family members or friends was associated with the highest predicted probability of vaccination willingness for health students (84.7%) and for non-health students (78.6%), while the negative recommendation of family members and friends was associated with the lowest predicted probability. Among the **influential voices** presented (Panel 4), the Minister of Health's recommendation was associated with the highest predicted probability for both groups. We observed that advice against vaccination was associated with reduced willingness. With regards to **vaccine protection**, vaccines with secondary protection were preferred by both groups (Panel 5). While the risk of **side effects** did not affect willingness for health students, everything except the referent parameter was associated with lower vaccination willingness for non-health students, with the lowest willingness associated with skin rash risk (Panel 6). We did formally compare vaccination willingness between groups. Upon testing whether the administration route affected students' attentiveness, we did not detect a significant difference between the groups (S1 Table).

## Discussion

This study explored the parameters that motivate willingness to receive a new vaccine against an emerging epidemic disease among university students in Kampala, Uganda. This study is unique in that it places disease context in the forefront, demonstrating how vaccination willingness varies by the proximity of the epidemic to the individual and severity of disease. We presented a general framing scenario reminiscent of Ebola outbreaks without naming any specific disease in order to assess participants' willingness to accept a newly developed vaccine in the context of a newly emergent disease. This scenario is particularly timely in light of the introduction of new vaccines against SARS-CoV-2.

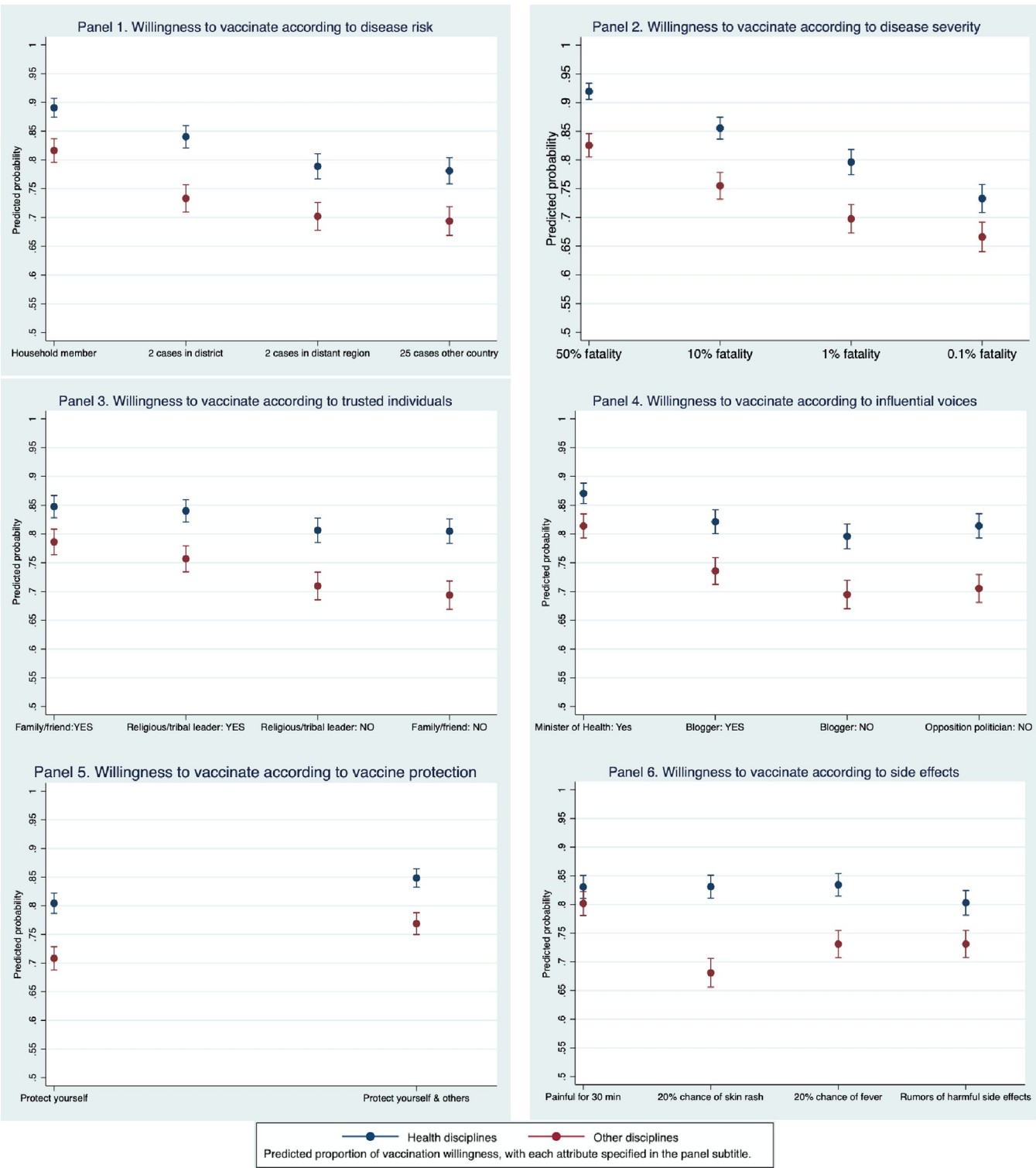

**Fig 2. Predicted probability and 95% CI of vaccination willingness for in six parameters: Disease risk, disease severity, trusted individuals, influential voices, vaccine protection and side effects.**

We found that vaccination willingness was high for students in both health disciplines and other disciplines, and that both groups maintained similar rankings of preferences between parameters presented. The strongest drivers of vaccine willingness were disease fatality rates and the proximity of infection risk and, among non-health students only, strongest negative impact from the risk of potentially serious side effect-a skin rash. The fact that the epidemiology of the disease—which included disease risk and disease severity- played the greatest role in willingness to accept vaccination among both groups is consistent with the positive relationship between pandemic risk and willingness to receive a vaccine documented in other settings [10,31,43].

We also found that vaccine recommendations or warnings had a positive, but smaller impact on willingness. Non-health students were more susceptible to negative messaging about vaccination and more deterred by the risk of side effects. The emergence of such suspicions may have major impact on a vaccination program, as seen in a DCE among French university students, where a controversy between a few health professionals and the Ministry of Health had the greatest absolute impact on vaccine acceptance [18]. We did not include any severe vaccine side effects in the DCE, as the ideal vaccine candidate would not be licensed if severe side effects were common.

Overall, we found that disease severity, the risk of contracting a disease, and the positive recommendation of the Minister of Health are the most important factors promoting acceptance of a newly developed vaccine to prevent against an emerging disease, and that side effects and warnings against vaccination are most strongly associated with lack of willingness. Planning for future vaccination campaign, such as the upcoming SARS-CoV-2 vaccine, should take these factors into account and begin raising vaccine awareness prior to vaccine availability to address concerns.

This study has several limitations. First, as a stated choice experiment, we cannot be certain that self-reported vaccine willingness would correlate with actual vaccination willingness in an epidemic context, a limitation common to all surveys of vaccine willingness (and stated preferences surveys) and potentially differential between interviewer and self-administered survey routes. To address this, staff adapted a standard DCE script to explain this potential bias towards willingness to students and encourage realistic answers [44], but we do not know whether this increased accuracy or addressed social desirability bias. We were unable to stratify the analysis by administration type as over 95% of those in health fields were permitted to choose their administration route; however, we did not detect a significant difference in internal validity between surveys with an administration choice and those which were only interview-administered in the sensitivity analysis.

Second, the study participants were drawn from a convenience sample of students, which is common in DCEs. This limitation could not be overcome because no registry was available to systemically invite eligible participants to the study. Thus, the results may not be representative of students in Uganda or of population-level willingness to receive a new vaccine. Thus, the results may not be representative of all students in Uganda, or the willingness of other subgroups to receive a new vaccine. This may be particularly relevant when the overlay of biological and social factors in an epidemic gives rise to increased susceptibility or worse outcomes for certain groups. Additional studies are needed to understand vaccination willingness in non-college populations. Although it was not possible to fully measure differential health-seeking behaviors between students in health disciplines and non-health disciplines, we sought to address some of these underlying differences by adjusting for Hepatitis B vaccination status. We did advertise widely and we observed that the demographics of the study are consistent with the student population by College and the distribution by sex for Makerere University [45], which is reassuring.

Despite these limitations, we designed and implemented a complex discrete choice experiment study, drawing a large and robust sample size and incorporating quality control measures. We undertook extensive efforts to collect high-quality data to answer critical questions about vaccine willingness using the most robust methods available. Our design increases internal validity and our results go beyond standard surveys to provide insight into the relative importance of drivers of vaccination willingness.

In conclusion, we undertook a discrete choice experiment to understand vaccination intent in the context of an epidemic of an emerging disease in Uganda. We found that vaccination willingness was greater when the epidemic was closer and the disease more severe. When considering new vaccine introduction in the context of a pandemic, policymakers should consider that vaccination willingness may shift over the course of an epidemic. Thus, vaccination campaigns that include messages that explain how quickly epidemic diseases can spread may help people to more adequately assess how their risk of disease may increase. We also found that vaccination willingness was influenced by who recommended vaccination. Especially during new vaccine introductions, trusted authority figures like the Minister of Health should publicly encourage vaccination and highlight herd immunity benefits. Policymakers should also proactively monitor and address rumors on social media and in the community as these could have a negative effect on vaccine willingness.

## Supporting information

**S1 Table. Sensitivity test by number, proportion, and predicted probability of difference in vaccination willingness between students in health disciplines and students in other disciplines.**
(DOCX)

**S2 Table. The odds ratio (OR) of attribute-specific parameters associated with willingness to receive a new vaccine, subset by sensitivity tests.**
(DOCX)

## Acknowledgments

We would like to acknowledge the research participants in this study as well as the NPGH Fogarty Global Health Fellowship and the Consortium for Law and Values in Health, Environment and the Life Sciences, University of Minnesota for funding this research.

## Author Contributions

**Conceptualization:** Kimberly E. Bonner, Judith E. Mueller, Traci Toomey, Cecily Banura, Nicole E. Basta.

**Data curation:** Kimberly E. Bonner.

**Formal analysis:** Kimberly E. Bonner, Jonathan Sicsic, Judith E. Mueller, Angela K. Ulrich, Keith J. Horvath.

**Funding acquisition:** Kimberly E. Bonner, Nicole E. Basta.

**Investigation:** Kimberly E. Bonner, Henry Ssekyanzi, Cecily Banura, Nicole E. Basta.

**Methodology:** Kimberly E. Bonner, Jonathan Sicsic, Judith E. Mueller, James D. Neaton, Cecily Banura, Nicole E. Basta.

**Project administration:** Kimberly E. Bonner, Henry Ssekyanzi, Cecily Banura, Nicole E. Basta.

**Resources:** Nicole E. Basta.

**Software:** Jonathan Sicsic, Nicole E. Basta.

**Supervision:** Judith E. Mueller, Traci Toomey, Keith J. Horvath, James D. Neaton, Cecily Banura, Nicole E. Basta.

**Validation:** Jonathan Sicsic.

**Visualization:** Kimberly E. Bonner.

**Writing – original draft:** Kimberly E. Bonner.

**Writing – review & editing:** Henry Ssekyanzi, Jonathan Sicsic, Judith E. Mueller, Traci Toomey, Angela K. Ulrich, Keith J. Horvath, James D. Neaton, Cecily Banura, Nicole E. Basta.

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
