## [Decision Letter · Decision Letter 0]

9 Sep 2021

PONE-D-21-15315

What drives willingness to receive a new vaccine that prevents an emerging infectious disease? A discrete choice experiment among university students in Uganda

PLOS ONE

Dear Dr. Bonner,

Thank you for submitting your manuscript to PLOS ONE. After careful consideration, we feel that it has merit but does not fully meet PLOS ONE’s publication criteria as it currently stands. Therefore, we invite you to submit a revised version of the manuscript that addresses the points raised during the review process.

Of note, I recruited reviewers with expertise in health decision making. Both were excited about the projects but expressed notable concerns that need to be addressed. Please pay particular attention to their concerns regarding the representiveness of the sample, choices made in composing the groups, and interpretation of the obtained differences

We look forward to receiving your revised manuscript.

Kind regards,

David P. Jarmolowicz, Ph.D.

Academic Editor

PLOS ONE

Journal Requirements:

2. Please ensure you have described how verbal consent was recorded and witnessed.

Reviewers' comments:

Reviewer's Responses to Questions

**Comments to the Author**

1. Is the manuscript technically sound, and do the data support the conclusions?

Reviewer #1: Yes

Reviewer #2: Yes

2. Has the statistical analysis been performed appropriately and rigorously? 

Reviewer #1: Yes

Reviewer #2: Yes

3. Have the authors made all data underlying the findings in their manuscript fully available?

Reviewer #1: Yes

Reviewer #2: No

4. Is the manuscript presented in an intelligible fashion and written in standard English?

Reviewer #1: Yes

Reviewer #2: Yes

5. Review Comments to the Author

Reviewer #1: Thank you for the opportunity to review this manuscript. Not only is this paper timely with its' content but addresses much needed information in the literature on vaccines and vaccination willingness

Reviewer #2: Thank you for the opportunity to review Bonner and colleagues’ manuscript titled “What drives willingness to receive a new vaccine that prevents an emerging infectious disease in Uganda.” The authors examined factors modulating the extent to which university students pursuing degrees in health versus other disciplines would be willing to receive a new vaccine if faced with an emerging infectious disease. Factors assessed included risk, severity, advise from trusted individuals, advise from influential voices (e.g., government; media), vaccine protection, and side effects. Students pursuing degrees in health disciplines reported significantly higher rates of willingness to receive the vaccine than those pursuing degrees in other disciplines. Of the potentially modulating factors considered, unsurprisingly the highest willingness was associated with conditions involving the greatest risks and severe outcomes. Advise and vaccine protection were most effective when coming from influential voices and preventing spread to others, respectively. Barriers included side effects and advice against vaccination, with non-health disciplines more susceptible to negative vaccine messaging and more deterred by the possibility of vaccine side effects. The document was well written, and the authors are commended for a timely investigation. Despite my enthusiasm for the work reported here, I must recommend that the manuscript be revised prior to further consideration for publication. Below are my primary concerns along with several suggestions that I believe will strengthen the paper and provide greater clarity for the reader.

1. The authors report that the sample consisted entirely of university students and, although it was acknowledged that these results may not generalize to “students in general in Uganda,” the authors make no mention of how these results may differ from population-level willingness to receive a new vaccine. Although the sample was drawn from different academic disciplines, it was nevertheless recruited from an institution of higher education, drawing the broader generality of the results into question. This should be discussed and acknowledged as a limitation.

2. Differences in willingness to receive a new vaccine between students in health and non-health disciplines might reflect more of a general tendency among those in health disciplines to engage in preventive measures. For example, the number of students in health disciplines to receive the Hepatitis B vaccine was more than double that of the non-health disciplines. If participants provided other information on health behaviors these should be reported and controlled for in the statistical models. If no such data were collected this should be addressed as a limitation.

3. Although the authors found no statistically significant differences in the number of participants excluded based on administration route (self- or interviewer-administered), it is unclear whether the authors examined differences in willingness to receive the vaccine across each of the key factors as a function of administration route. The likelihood of reactivity when answering interviewer-administered questions about the potential for life and death should be addressed either by clearly reporting any differences in willingness between administration methods or acknowledging this as a limitation.

4. Although dichotomizing the sample into health versus “other” disciplines resulted in notable differences in willingness to receive a new vaccine, it may have hidden valuable information. Specifically, aggregating all non-health disciplines into an “other” category precluded identification of disciplines that may have indicated levels of willingness comparable to the health discipline. Consider including a supplementary table that provides descriptive data on willingness by specific discipline (or college) or address this point in the discussion.

5. Please confirm the accuracy of the results of the statistical analyses and the data reported in the tables.

Minor comments:

Consider combining the second and third study aims by saying “…to identify the parameters that modulate vaccination willingness…” rather than differentiating vaccination willingness from motivating parameters.

The authors report conducting a literature review to identify 6 key attributes that may influence willingness to receive a new vaccine. It is unclear based on this language whether the goal was to identify 6 key attributes at the outset or if, based on the literature, 6 key attributes emerged. Please adjust the language accordingly to provide greater clarity for the reader.

Related to the above point, if there was a systematic method of identifying the key attributes (e.g., frequency within the literature; magnitude of associations) please report this. If not please briefly provide some information on why these attributes were chosen.

Supplementary Table 1:

For Item 4 (Administration) the number of health disciplines excluded is reported as n = 775 (99%). Please provide the correct n(%). It is also unclear based on this table how many participants assigned to each administration method were excluded for failing the “Duplicates” sensitivity tests. Please provide this information in the table.

6. PLOS authors have the option to publish the peer review history of their article (what does this mean?). If published, this will include your full peer review and any attached files.

Reviewer #1: No

Reviewer #2: No

---

## [Author Response · Author response to Decision Letter 0]

1 Nov 2021

Response to reviewers: PONE-D-21-15315 

What drives willingness to receive a new vaccine that prevents an emerging infectious disease? A discrete choice experiment among university students in Uganda 

Dear Reviewers,

Thank you for your thoughtful review of our manuscript entitled, “What drives willingness to receive a new vaccine that prevents an emerging infectious disease? A discrete choice experiment among university students in Uganda”. We appreciate your suggestions and have undertaken extensive efforts to fully address each comment. 

In the section following, we provide a listing of each reviewer comments and a detailed description of the ways we have addressed these comments and the changes we have made to the manuscript. Reviewer text is italicized for clarity.

Reviewer 1 Suggestions and Responses

1. The authors report that the sample consisted entirely of university students and, although it was acknowledged that these results may not generalize to “students in general in Uganda,” the authors make no mention of how these results may differ from population-level willingness to receive a new vaccine. Although the sample was drawn from different academic disciplines, it was nevertheless recruited from an institution of higher education, drawing the broader generality of the results into question. This should be discussed and acknowledged as a limitation.

 Thank you for raising this point. We have broadened this limitation to address the mention of how these results may differ from population-level willingness to receive a new vaccine on lines 400-401:

“Thus, the results may not be representative of students in Uganda or of population-level willingness to receive a new vaccine. Additional studies are needed to understand vaccination willingness in non-college populations.”

2. Differences in willingness to receive a new vaccine between students in health and non-health disciplines might reflect more of a general tendency among those in health disciplines to engage in preventive measures. For example, the number of students in health disciplines to receive the Hepatitis B vaccine was more than double that of the non-health disciplines. If participants provided other information on health behaviors these should be reported and controlled for in the statistical models. If no such data were collected this should be addressed as a limitation.

Health seeking behaviors overall may differ between students in the health sciences and students outside the health sciences. We adjusted for Hepatitis B vaccination status as a proxy for these health seeking behaviors. However, as our research questions do not ascertain other health behaviors or the relative contribution of differential uptake of health behaviors, it was not possible to address or investigate reasons for differences between these groups.

We have added the following text to the limitations, line 419-422:

“Although it was not possible to fully measure differential health-seeking behaviors between students in health disciplines and non-health disciplines, we sought to address some of these underlying differences by adjusting for Hepatitis B vaccination status.”

3. Although the authors found no statistically significant differences in the number of participants excluded based on administration route (self- or interviewer-administered), it is unclear whether the authors examined differences in willingness to receive the vaccine across each of the key factors as a function of administration route. The likelihood of reactivity when answering interviewer-administered questions about the potential for life and death should be addressed either by clearly reporting any differences in willingness between administration methods or acknowledging this as a limitation.

 In the sensitivity analyses, we did not detect a significant difference in internal validity between surveys with an administration choice and those which were only interview-administered.

The potential for differential social desirability bias between these two administration routes is certainly possible. Given the differential distribution of administration routes by discipline, we have added the following text in the limitation to note this issue in lines 391-395:

“First, as a stated choice experiment, we cannot be certain that self-reported vaccine willingness would correlate with actual vaccination willingness in an epidemic context, a limitation common to all surveys of vaccine willingness (and stated preferences surveys) and potentially differential between interviewer and self-administered survey routes. To address this, staff adapted a standard DCE script to explain this potential bias towards willingness to students and encourage realistic answers (40), but we do not know whether this increased accuracy or addressed social desirability bias. We were unable to stratify the analysis by administration type as over 95% of those in health fields were permitted to choose their administration route; however, we did not detect a significant difference in internal validity between surveys with an administration choice and those which were only interview-administered in the sensitivity analysis.”

4. Although dichotomizing the sample into health versus “other” disciplines resulted in notable differences in willingness to receive a new vaccine, it may have hidden valuable information. Specifically, aggregating all non-health disciplines into an “other” category precluded identification of disciplines that may have indicated levels of willingness comparable to the health discipline. Consider including a supplementary table that provides descriptive data on willingness by specific discipline (or college) or address this point in the discussion.

In designing this study, we planned our analyses to include a comparison of health disciplines vs all other disciplines a priori. We powered the study to ensure that this comparison would have an adequate sample size and lead to robust inference. Post-hoc subgroup analyses are not advisable in epidemiologic research given the likelihood of identifying spurious associations that do not reflect the true associations of interest. Future studies could investigate differences in willingness by subfield by establishing this is a questions of interest at the outset of the study and ensuring the study is adequately powered to answer these questions.   5. Please confirm the accuracy of the results of the statistical analyses and the data reported in the tables. 

Thank you-we have confirmed the accuracy of the results of these analyses and data reported in tables.

6. Consider combining the second and third study aims by saying “…to identify the parameters that modulate vaccination willingness…” rather than differentiating vaccination willingness from motivating parameters. 

In Aim 2, we aim to identify which parameters among those investigated drive willingness to be vaccinated overall and among students in health fields and all others disciplines, whereas in Aim 3, our goal is to compare the magnitude of the association between those parameters identified in Aim 2 and willingness to be vaccinated overall and among students in health fields and all others disciplines. 

To address these distinct questions, we developed two different analysis models. The model used to address aim 2 differs from the model to address 3. We have made edits to the Aims to clarify. Both of these analyses are important because we are seeking to 1) identify which factors drive this willingness, and to 2) assess the relative strength of those factors in driving willingness to be vaccinated.

We have added a clarifying sentence to the specific aims description in lines 146-157:

“We had three primary aims: (1) to estimate and compare willingness to be vaccinated among university students in health fields (those enrolled in the College of Health Sciences) and students enrolled in other fields; (2) to identify which parameters drive willingness to be vaccinated overall and among students in health fields and all others disciplines; and (3) to compare the magnitude of the association between those parameters identified in Aim 2 and willingness to be vaccinated overall and among students in health fields and all others disciplines. By accomplishing these aims, we will be able to describe willingness to be vaccinated with a new vaccine for an emerging disease among university students, to identify which factors drive this willingness, and to assess the relative strength of those factors in driving willingness to be vaccinated. Our goal is to identify factors that are most strongly associated with willingness so that future vaccination campaigns can address these factors when new vaccines are introduced. ”

 7. The authors report conducting a literature review to identify 6 key attributes that may influence willingness to receive a new vaccine. It is unclear based on this language whether the goal was to identify 6 key attributes at the outset or if, based on the literature, 6 key attributes emerged. Please adjust the language accordingly to provide greater clarity for the reader. 

Related to the above point, if there was a systematic method of identifying the key attributes (e.g., frequency within the literature; magnitude of associations) please report this. If not please briefly provide some information on why these attributes were chosen.

 To identify key attributes, we followed the International Society for Pharmacoeconomics Outcomes Research (ISPOR) guidelines to identify attributes for this analysis(see excerpt full citation following). Systematic reviews are not the standard for attribute selection.

Bridges JF, Hauber AB, Marshall D, Lloyd A, Prosser LA, Regier DA, et al. Conjoint analysis applications in health--a checklist: a report of the ISPOR Good Research Practices for Conjoint Analysis Task Force. Value Health. 2011;14(4):403-13.

We have clarified the introduction to illustrate that these attributes emerged through a review of the literature in lines 227-231. 

“We undertook a literature review of other DCEs on adult vaccination (8, 16, 17, 27, 34, 35) and the broader literature on vaccine willingness, which identified six key attributes that could influence vaccination willingness in the context of an emerging disease. We listed the attributes identified by the literature review and selected the six attributes with the greatest magnitude of effect.”

Supplementary Table 1:  9. For Item 4 (Administration) the number of health disciplines excluded is reported as n = 775 (99%). Please provide the correct n(%). It is also unclear based on this table how many participants assigned to each administration method were excluded for failing the “Duplicates” sensitivity tests. Please provide this information in the table.

Supplementary Table 1 shows the number and percentage of students that met the criteria for each of the sensitivity tests; the number of excluded students from each group; the risk difference in willingness to accept a vaccine between groups; and the risk difference in answering consistently answering the duplicate questions between groups. Item 4 examines the proportion of students consistently responding to the duplicate survey questions for each administration route. The value reported is correct. In this analysis, we aimed to restrict each group to those who consistently respondent to the duplicate questions as a proxy for participant focus on the survey, and then assess if willingness to vaccinate changed in each group when each group was restricted to these subsets. Since nearly all participants (775/783) were in the health sciences group, there weren't enough participants to assess differences in willingness to vaccinate between surveys.

Reviewer 2 Suggestions and Responses

1. Some clarity between vaccination willingness and motivating parameters would be of benefit to understand the difference of and need for Aims 2 and 3. In the paper, the authors note that the WHO incorporates vaccination willingness into vaccination motivation, however, the paper has them separated out into two separate aims. Are these two separate components that need to be investigated separately, or like the WHO notes, are these one in the same? If they are one in the same, please give some rationale as to why they need to be explored as separate processes. Some clarity as to this decision would help clear up some confusion to the reader.

The WHO’s behavioral and social drivers (BeSD) for vaccination framework (below) identifies 1. what people think and feel, 2. social norms, 3. motivation, and 4. access as the key factors that influence vaccination uptake. While item 3 "motivation" is a factor influencing vaccination uptake, a number of other factors could influence an individual's motivation.

In this study, we are seeking to understand both:

1) Which factors influence willingness to be vaccinated (as a proxy for vaccination uptake) and

2) The extent to which these factors- defined as parameters within the manuscript, in keeping with the Discrete Choice Experiment literature- within the BeSD framework drive vaccination intent overall and among students in health fields and other disciplines. 

These questions are similar, but distinct, and they require separate models. We included both because it is important to know both the factors that drive vaccination intent and the extent to which these factors drive vaccination intent. The exact pathway between vaccination and uptake is beyond the scope of this paper, as this paper explored a scenario of different vaccine profiles and epidemic contexts for a newly-emerging epidemic disease.

The text has been updated to reflect the relationship between motivating parameters and vaccination willingness, using the BeSD framework in line 146-157:

“…We had three primary aims: (1) to estimate and compare willingness to be vaccinated among university students in health fields (those enrolled in the College of Health Sciences) and students enrolled in other fields; (2) to identify which parameters drive willingness to be vaccinated overall and among students in health fields and all others disciplines; and (3) to compare the magnitude of the association between those parameters identified in Aim 2 and willingness to be vaccinated overall and among students in health fields and all others disciplines. By accomplishing these aims, we will be able to describe willingness to be vaccinated with a new vaccine for an emerging disease among university students, to identify which factors drive this willingness, and to assess the relative strength of those factors in driving willingness to be vaccinated. Our goal is to identify factors that are most strongly associated with willingness so that future vaccination campaigns can address these factors when new vaccines are introduced.”

2. Aim 1 is stated as comparing the overall differences in health professionals and other university students in Uganda. In the study design, the authors note that participants were recruited from the 6 largest colleges at Makerere University, yet the way the schools are listed it is difficult to determine if there are 6 or 8 listed. Using a numbered list might help to separate the schools so it is more apparent due to the multiple commas and “and’s” listed. Relatedly, there were six schools but only five collection sites. Were all school located equal distant from these collection sites? A single sentence added to address this question would be of benefit to the methods section. 

Thank you. Numbers have been added to lines 156-160:

“Eligible students were aged 18 years and above, able to read and speak English (the national language of Uganda) and were current students in one of the six largest Colleges at Makerere University (1. Business and Management Sciences, 2. Computing and Information Science, 3. Education and External Studies, 4. Health Science, 5. Humanities and Social Science, and 6. Veterinary Medicine, Animal Resources, and Biosecurity).”

We now note in the methods lines 169-195, that pilot testing occurred at one data collection site in one school and survey data collection occurred at five sites distributed among the other schools: 

“Pilot testing occurred at the College of Veterinary Medicine prior to the launch of the study. For the survey, a convenience sample was recruited through posters and WhatsApp messages sent by student leaders. Participants were enrolled from all six colleges across five enrollment sites.”

3. It would be of use to explain which are considered health professionals within the paper. Is health science the only school considered health professionals or are veterinary medicine students lumped into this? My understanding as a reader is that the authors are evaluating those students with some background training in medicine and the body which would seem to include veterinary medicine, but this is only my assumption. Some clarity would be beneficial to understanding what metrics were used to determine health professionals. If veterinary medicine is not included in this, it would be helpful to understand how this might impact the results for the non-health professional groups.

Eligible students from the College of Veterinary Medicine participated in the survey pilot-testing, but were excluded from main study [survey itself]. We defined " future health professionals" for the purposes of this analysis as those students enrolled in the "College of Health Sciences" only. We made this distinction because we are interested in the assessing willingness among students training to become human health professionals who may themselves administer vaccines to patients or advise patients on matters of vaccination. In previous studies, clinicians' willingness to get vaccinated themselves has been associated with their willingness to advise patients to get vaccinated or to vaccinate their children. 

We clarify in lines 109-111 in the background: 

“Here we define health fields as those in which students might themselves administer vaccines to patients or advise patients on matters of vaccination.”

And we clarify in lines 168-169 of the methods:

“We define students in health fields as those enrolled in the College of Health Sciences.”

4. Additionally, overall clarity as to “health professionals,” within the introduction should be addressed. Information about and citations for “health professionals” are used throughout the introduction, yet the study was conducted on students. The authors state that “health professionals play a critical role in promoting vaccination. Therefore, there is a need to understand what motivates health professional’s willingness as their personal…” Are the students considered health professionals? As a reader, I am confused as to the classification of the students as students or professionals and if this is the same for the citations used to reference health professionals. 

Thank you for raising this point. Our aim in this study was to assess vaccine willingness among students training to become health professionals because studies have shown that once health professionals are practicing, their views on vaccination are important drivers: We have added the following text in lines 107-111:

“Additionally, there is a need to understand the willingness of those who have newly entered or will soon enter the health workforce, as these health professionals can influence patients throughout the course of their careers.” 

5. Throughout the introduction, vaccination willingness is mentioned but never defined or explained for the purpose of this study. Even as the authors’ note that there is a growing body of research on vaccination willingness, this topic is not further elucidated for the reader until the end of the introduction. A brief introduction into the components of vaccination willingness earlier in the introduction would do well to help the reader understand the subtleties of this term. This is clarified with the paragraph on key attributes for the study, but it comes after all the information pertinent to this topic. Even a short list of these earlier in the paper could help add some clarity. 

Thanks for raising this. Vaccination willingness is defined as having an intent or motivation to be vaccinated and is used as an indicator for possible future vaccination uptake. We have amended the introduction: lines 70-72 to define this concept from the beginning.

“However, the impact of any newly developed vaccine depends on the proportion of individuals who express willingness to be vaccinated and seek out vaccination. Willingness is defined as having an intent or motivation to be vaccinated in the future with a hypothetical novel vaccine.”

Minor edits:

6. The paper would benefit from some clarification as to why/how the citations for studies from different countries contribute to this specific study. Specifically, the authors note that for the Ebola vaccine acceptability was high in Nigeria and Sierra Leone, but very low in the US. Does this reflect a lower willingness in more developed countries? If so, why would this study focus on Uganda? Additionally, the authors note that Liberia had a low acceptance rate of the Ebola vaccine by healthcare works. Yet, they note that the highest vaccine hesitancy took place in Uganda when immunization managers were surveyed. Further, they cite results from a study done in France that identifies key drivers of vaccination willingness. Some clarification of these competing results would make this paper much more impactful, specifically, addressing the difference in study results for developed versus developing countries.

Thank you for raising this important point. We reviewed the literature to identify any studies that assessed vaccination intent for an emerging epidemic disease in any context. Given the very limited number of studies that have addressed this question in any context, we provided a summary of all studies that met these criteria. We focused specifically on Ebola vaccine because Ebola is an emerging epidemic disease for which a new vaccine has been developed and disease which has been detected in Uganda in at least five separate outbreaks. Vaccination willingness has been examined in 67 countries using a four-question assessment, but was only asked in four countries in sub-Saharan Africa. In an assessment of Immunization Program managers in 13 countries globally, Uganda was identified as vaccine hesitancy playing the largest role in hindering vaccine uptake. 

An unpublished systematic review [citation below] from 2016 found that 80% vaccine-focused discrete choice experiments were conducted in high income countries, whereas only one vaccine-focused DCE [Verelst et al] had been undertaken in any country in sub-Saharan Africa. 

Verelst et al sought to elucidate the social norms that drive vaccination intent in South Africa, including population and local vaccination coverage. We drew from attributes regarding the disease itself and side effects. Seanehia et al examine vaccination intent amongst university students in France, and we drew all of their attributes identified:  epidemic situation, adverse events, communication regarding the vaccine, and potential for indirect protection. Additionally, we drew from the attributes identified by Determann et al which examined drivers of vaccination intent towards a pandemic vaccine amongst adults in Belgium, including disease characteristics, media attention, and vaccine safety. 

As DCEs are being increasingly undertaken in public health and vaccination research, we hope that future studies will be able to draw from robust data sources within countries and populations.

Poulos C. A review of discrete choice experiment studies of preferences for vaccine features. Poster presented at the 2016 ISPOR 21st Annual International Meeting; May 24, 2016. Washington, DC. [abstract] Value Health. 2016 May; 19(3):A220.

Piecemeal 

7. Within this same framework, it would be useful for the authors to address the generalizability of these results to the larger population of Uganda, or even more widely, to the world. Around 75% of Uganda residents live on less than $2 a day. Is the sample population relevant to the larger population? Does surveying those at university inform vaccination willingness for the broader scope of residents of the country? Either way this topic would be important to note for readers and researchers. It seems like comparing the college population to the non-college population might have differences that are reflective of evaluating health professionals versus non-health professionals

This study focuses on future health professionals in Uganda because prior research has demonstrated the influential role that health professionals have on promoting vaccination uptake. As such, we sought to understand attitudes towards vaccination among students, comparing future health professionals with other students to give us insight into how the next generation of health professionals will view vaccinations against emerging infectious diseases, vaccinations that will inevitably be developing during their careers. 

We have noted this aim in line 107-111 and have stated in the limitations (lines 421-426) that the results are not intended to generalize to the general population of Uganda and the world. 

Lines 107-111:

Additionally, there is a need to understand the willingness of those who have newly entered or will soon enter the health workforce, as these health professionals can influence patients throughout the course of their careers. Here we define health fields as those in which students might themselves administer vaccines to patients or advise patients on matters of vaccination.

Lines 421-426:

Thus, the results may not be representative of students in Uganda or of population-level willingness to receive a new vaccine. Additional studies are needed to understand vaccination willingness in non-college populations. Although it was not possible to fully measure differential health-seeking behaviors between students in health disciplines and non-health disciplines, we sought to address some of these underlying differences by adjusting for Hepatitis B vaccination status. 

8. The citations used for other country vaccination willingness was mostly looking at the general population, however, this study exclusively evaluated educated and therefore mostly likely higher SES samples. Does this influence the results or how it may generalize to the overall population in Uganda or other countries?

In the discussion, we have cited seven DCEs that have addressed factors associated with vaccine uptake. Among those, only one (Verelst et. Al.) study was conducted in sub-Saharan Africa, highlighting the exceptional limited use of this novel methodology in low and middle income countries. We intentionally sampled future health professionals and compared their views with other students.

The aim of our study was not to characterize vaccine acceptance among the general population in Uganda or other countries, but to assess the views of future health professionals and compare their views to those of their peers. We clarify and emphasize these points in the limitations section to ensure that readers do not attempt to generalize to the general population. Future studies are needed in the general population and we hope that our study will serve as a model for how to undertake such a study. 

9. In the results, there are certain parameters, such as Influential voices, that increase willingness for non-health professionals more so than the health professionals. The overall result was reported, but information as to why this was seen could be of interest to researchers. Does background/education in health sciences decrease the importance of influential voices on the willingness to take a vaccine? Though it is at odds with the other findings, this is actually a very informative finding when generalizing to the lower educated and lower SES groups of Uganda. 

Thanks for raising this point. Because baseline vaccination willingness was lower for participants in non-health disciplines compared to the College of Health Sciences, a parameter may be associated with higher odds of vaccination willingness, but an overall lower predicted probability of vaccination willingness. Thus, the students in the College of Health Sciences maintained a higher predicted probability of vaccination across allof the parameter options for the “influential voices” attribute.

Of note is the relatively low vaccination willingness associated with negative vaccination recommendations for participants in the non-health disciplines. These findings suggest that misinformation or disinformation can play a larger role in reducing vaccination willingness for students trained in disciplines other than health sciences. However, as noted above, the study is not designed to allow for the results to be generalized beyond an educated population of young adults and we have strengthened this point in the limitation section to clarify. However, these are important questions that we hope to explore in future research.

10. In the paragraph of lines 170-174, the authors note that two data collection sites allowed participants to choose between self-administering the survey and having a staff member administer it. At three additional sites it was only allowed to be administered by staff. Although the authors analyzed and reported results for this difference, it might be advantageous address why both sites did not allow for this option. 

Thanks for raising this. When the study launched, we gave participants the option for self-administration or interviewer survey administration. Because the self-administered surveys were being completed much more quickly than the interviewer-administered and because we wished to ensure the highest level of data quality and data completeness as possible. This is why some sites allowed an option for administration route and some sites allowed only interviewer-administration. In the sensitivity analyses, we did not detect a significant difference in internal validity between surveys with an administration choice and those which were only interview-administered, and thus, the shift to only interviewer-administered surveys may not have been necessary. We have added note of social desirability bias in the limitations sections, lines 373-379:

 “First, as a stated choice experiment, we cannot be certain that self-reported vaccine willingness would correlate with actual vaccination willingness in an epidemic context, a limitation common to all surveys of vaccine willingness (and stated preferences surveys) and potentially differential between interviewer and self-administered survey routes. To address this, staff adapted a standard DCE script to explain this potential bias towards willingness to students and encourage realistic answers (40), but we do not know whether this increased accuracy or addressed social desirability bias.”

Grammar and Spelling:

Line 133 should state “study” instead of “atudy”

Line 321 states that “This scenario is particularly timely in light of the anticipated introduction of new vaccines against SARS-CoV-2.” The authors might consider amending this to “timely in light of the introduction of…” since multiple vaccines are already being administered worldwide.

Thank you for identifying these issues. They have been addressed in the manuscript.

Small items and personal feedback for the authors (These are items that do not necessarily need to be addressed, but I wanted to convey to the authors as a personal note):

Some paragraphs are rather short, however, almost half of the paragraph is one sentence. It may do well to help the reader by cutting down some of these longer sentences to make the information more digestable. The overcomplexity of these sentences could have a negative impact on the ability of the reader to absorb and retain all the relevant points. 

Although not specific to epidemic vaccination and DCE’s there is a related literature that would be of note for the authors to review or at least be aware of. Listed below are a few papers that directly investigate multiple influences on medication willingness and adherence by manipulating multiple variables related to efficacy, side effects and likelihood of taking a medication. These papers, among others in related literature, would help to support and explain some of the information that is left unexplained in the introduction. Again, this is not directly related to DCE’s or vaccines, but is related to manipulation of multiple variables in a pro-health context.

Bruce, J. M., Bruce, A. S., Catley, D., Lynch, S., Goggin, K., Reed, D., Jarmolowicz, D. P. (2016). Being kind to your future self: Probability discounting of health decision-making. Annals of Behavioral Medicine, 50, 297–309. http://dx.doi.org/10.1007/s12160-015-9754-8

Jarmolowcz, D. P., Reed, D. D., Bruce, A. S., Lynch, S., Smith, J., Bruce, J. M. (2018). Modeling effects of side effect probability, side-effect severity, and medication efficacy on patients with multiple sclerosis medication choice. Experimental and Clinical Psychopharmacology. 26 (6), 599-607. http://dx.doi.org/10.1037/pha0000220

Jarmolowicz, D. P., Reed, D. D., Schneider, T. D., Smith, J., Thelen, J., Lynch, S., Bruce, A. S., & Bruce, J. M. (2019). Behavioral economic demand for medications and its relation to clinical measures in Multiple Sclerosis. Experimental and Clinical Psychopharmacology.

Thank you for highlighting these studies-it is always helpful to gain insight from complementary literature on these important aspects of behavior and choice. We have highlighted literature relevant to vaccination in this manuscript to align with the aims of the study.

---

## [Decision Letter · Decision Letter 1]

30 Dec 2021

PONE-D-21-15315R1What drives willingness to receive a new vaccine that prevents an emerging infectious disease? A discrete choice experiment among university students in UgandaPLOS ONE

Dear Dr. Bonner,

Thank you for submitting your manuscript to PLOS ONE. After careful consideration, we feel that it has merit but does not fully meet PLOS ONE’s publication criteria as it currently stands. Therefore, we invite you to submit a revised version of the manuscript that addresses the points raised during the review process.

We look forward to receiving your revised manuscript.

Kind regards,

David P. Jarmolowicz, Ph.D.

Academic Editor

PLOS ONE

Journal Requirements:

Reviewers' comments:

Reviewer's Responses to Questions

**Comments to the Author**

1. If the authors have adequately addressed your comments raised in a previous round of review and you feel that this manuscript is now acceptable for publication, you may indicate that here to bypass the “Comments to the Author” section, enter your conflict of interest statement in the “Confidential to Editor” section, and submit your "Accept" recommendation.

Reviewer #2: All comments have been addressed

Reviewer #3: (No Response)

2. Is the manuscript technically sound, and do the data support the conclusions?

Reviewer #2: (No Response)

Reviewer #3: Yes

3. Has the statistical analysis been performed appropriately and rigorously? 

Reviewer #2: (No Response)

Reviewer #3: Yes

4. Have the authors made all data underlying the findings in their manuscript fully available?

Reviewer #2: (No Response)

Reviewer #3: Yes

5. Is the manuscript presented in an intelligible fashion and written in standard English?

Reviewer #2: (No Response)

Reviewer #3: Yes

6. Review Comments to the Author

Reviewer #2: (No Response)

Reviewer #3: The authors did a great job of improving this manuscript based on earlier reviewer feedback. One item remains to be addressed, which may or may not have been part of the earlier feedback from reviewers. The authors do not provide adequate theoretical context for their work. I would have appreciated some reference to the Health Belief model (see https://sphweb.bumc.bu.edu/otlt/mph-modules/sb/behavioralchangetheories/behavioralchangetheories2.html for an overview), as well as the articles on delay discounting/behavioral economics provided in the previous editorial feedback, since these concepts are directly relevant to the manuscript and this research. Finally, I would like to point out that the additional concept of Syndemics might be much more appropriate as a framing, rather than referring to epidemic as the context. A syndemic is the interaction between multiple epidemics and includes behavioral and political context. Persons make choices within those contexts, so they should not be ignored in the framing of this research.

7. PLOS authors have the option to publish the peer review history of their article (what does this mean?). If published, this will include your full peer review and any attached files.

Reviewer #2: No

Reviewer #3: No

---

## [Author Response · Author response to Decision Letter 1]

2 Feb 2022

Dear Reviewers,

Thank you for your thoughtful review of our manuscript entitled, “What drives willingness to receive a new vaccine that prevents an emerging infectious disease? A discrete choice experiment among university students in Uganda”. We appreciate your suggestions and have undertaken extensive efforts to fully address each comment. 

In the section following, we provide a listing of each reviewer comments and a detailed description of the ways we have addressed these comments and the changes we have made to the manuscript. Reviewer text is italicized for clarity.

Reviewer 3 Suggestions and Responses

1. The authors did a great job of improving this manuscript based on earlier reviewer feedback. One item remains to be addressed, which may or may not have been part of the earlier feedback from reviewers. The authors do not provide adequate theoretical context for their work. I would have appreciated some reference to the Health Belief model (see https://sphweb.bumc.bu.edu/otlt/mph-modules/sb/behavioralchangetheories/behavioralchangetheories2.html for an overview), as well as the articles on delay discounting/behavioral economics provided in the previous editorial feedback, since these concepts are directly relevant to the manuscript and this research. 

Thank you for your positive feedback about our prior revisions. We chose to use the WHO’s behavioral and social drivers (BeSD) for vaccination framework (below), based upon the Increasing Vaccination Model, and largely derived from psychology research. We had considered the Health Belief model when developing this research. However, the Health Belief Model only captures certain elements that are outlined in the BeSD framework: the thinking and feeling domain, specifically the risk appraisal and vaccine confidence constructs. Our discussion of the BeSD and the Increasing Vaccination Model encompassess the concepts of the Health Believe Model and incorporates additional concepts that are also critically relevant to our study. To make this clear, we have noted this in the manuscript text and added a specific reference to the Health Belief model in lines 74-77:

“This [Increasing Vaccination] model incorporates the health belief model, including risk appraisal and vaccine confidence, into the thinking and feeling domain, while adding additional domains, including social process, and practical issues (2, 3).” 

We agree that DCEs in general and our study in particular are grounded in behavioural economics theory, which can help explaining vaccination refusal, due to risk aversion (of side effects), over-weighting of probabilities of rare events (e.g., serious sides effects), status quo bias, or preference for the present (high discounting of future immunity).

However, the concept of delay discounting does not explicitly enter our study, as we are examining trade-offs made for willingness to accept, but not willingness to pay. In our study, we recognized that a vaccine for an emerging infectious disease in Uganda would almost certainly be made available free of charge as are other vaccines recommended in the country.  Discounting would find its place in modelling to identify optimized strategies.

To raise the importance of concepts from behavioral economics and clarify which elements we considered in designing this study with the aims presented given the cross-disciplinary nature of our study, we now note the importance of behavioural economics. The references suggested by the reviewer do not refer to concepts relevant to our study. Instead, we have identified two citations that illustrate the key points and have included those in the text lines 125-128:

“DCEs measure the dominant driver of a single decision when there are multiple competing factors. Some concepts of behavioral economics, such as discounting, are explicited in DCE only if economic considerations are central to the aims of the study (26, 27), but not in a DCE on willingness to accept a vaccine.”

2. Finally, I would like to point out that the additional concept of Syndemics might be much more appropriate as a framing, rather than referring to epidemic as the context. A syndemic is the interaction between multiple epidemics and includes behavioral and political context. Persons make choices within those contexts, so they should not be ignored in the framing of this research.

Thanks for raising the question as to whether syndemic, a term which has gained much greater awareness during the COVID-19 pandemic, might be more relevant than epidemic to our study. As noted by Tsai et al (Lancet 2017), "As originally theorised, three concepts underlie the notion of a syndemic: disease concentration, disease interaction, and the large-scale social forces that give rise to them." We framed our study in terms of an epidemic because this is how the scenarios were presented to participants. We aimed to examine how participants would individually respond in the setting of a specific, acute disease outbreak-an epidemic; we did not incorporate how such an epidemic might interact with other diseases or with social and environmental factors or how these factors might shape disease dynamics within this context. Given our aims, the focus of our study on a single disease, and the emphasis we placed on examining decision-making for vaccination given the epidemiologic context alone, we believe that epidemic more accurately describes the nature of the context/scenarios presented in the study. However, we have made note of the utility of considering the broader context in future studies in the Discussion section, lines 386-389:

“Thus, the results may not be representative of all students in Uganda, or the willingness of other subgroups to receive a new vaccine. This may be particularly relevant when the overlay of biological and social factors in an epidemic gives rise to increased susceptibility or worse outcomes for certain groups.”

---

## [Decision Letter · Decision Letter 2]

22 Apr 2022

What drives willingness to receive a new vaccine that prevents an emerging infectious disease? A discrete choice experiment among university students in Uganda

PONE-D-21-15315R2

Dear Dr. Bonner,

We’re pleased to inform you that your manuscript has been judged scientifically suitable for publication and will be formally accepted for publication once it meets all outstanding technical requirements.

Kind regards,

Iván Barreda-Tarrazona, PhD

Academic Editor

PLOS ONE

Additional Editor Comments (optional):

Reviewers' comments:

Reviewer's Responses to Questions

**Comments to the Author**

1. If the authors have adequately addressed your comments raised in a previous round of review and you feel that this manuscript is now acceptable for publication, you may indicate that here to bypass the “Comments to the Author” section, enter your conflict of interest statement in the “Confidential to Editor” section, and submit your "Accept" recommendation.

Reviewer #3: All comments have been addressed

2. Is the manuscript technically sound, and do the data support the conclusions?

Reviewer #3: Yes

3. Has the statistical analysis been performed appropriately and rigorously? 

Reviewer #3: Yes

4. Have the authors made all data underlying the findings in their manuscript fully available?

Reviewer #3: Yes

5. Is the manuscript presented in an intelligible fashion and written in standard English?

Reviewer #3: Yes

6. Review Comments to the Author

Reviewer #3: Thank you for the thoughtful consideration of editorial comments, and for making improvements to the manuscript.

7. PLOS authors have the option to publish the peer review history of their article (what does this mean?). If published, this will include your full peer review and any attached files.

Reviewer #3: No

---

## [Editor Report · Acceptance letter]

12 May 2022

PONE-D-21-15315R2 

What drives willingness to receive a new vaccine that prevents an emerging infectious disease? A discrete choice experiment among university students in Uganda 

Dear Dr. Bonner:

I'm pleased to inform you that your manuscript has been deemed suitable for publication in PLOS ONE. Congratulations! Your manuscript is now with our production department. 

Kind regards, 

on behalf of

Dr. Iván Barreda-Tarrazona 

Academic Editor

PLOS ONE